# Arylpiperazine Derivatives and Cancer: A New Challenge in Medicinal Chemistry

**DOI:** 10.3390/ph17101320

**Published:** 2024-10-02

**Authors:** Giorgia Andreozzi, Angela Corvino, Beatrice Severino, Elisa Magli, Elisa Perissutti, Francesco Frecentese, Vincenzo Santagada, Giuseppe Caliendo, Ferdinando Fiorino

**Affiliations:** 1Dipartimento di Farmacia, Università di Napoli Federico II, Via D. Montesano 49, 80131 Naples, Italy; giorgia.andreozzi@unina.it (G.A.); angela.corvino@unina.it (A.C.); bseverin@unina.it (B.S.); elisa.magli@unina.it (E.M.); perissut@unina.it (E.P.); frecente@unina.it (F.F.); santagad@unina.it (V.S.); caliendo@unina.it (G.C.); 2Dipartimento di Sanità Pubblica, Università di Napoli Federico II, Via Pansini 5, 80131 Naples, Italy

**Keywords:** arylpiperazine, cancer, small molecules, anti-proliferative agents

## Abstract

**Background:** In recent decades, there has been a startling rise in the number of cancer patients worldwide, which has led to an amazing upsurge in the development of novel anticancer treatment candidates. On a positive note, arylpiperazines have garnered attention in cancer research due to their potential as scaffolds for developing anticancer agents. These compounds exhibit a diverse array of biological activities, including cytotoxic effects against cancer cells. Indeed, one of the key advantages of arylpiperazines lies in their ability to interact with various molecular targets implicated in cancer pathogenesis. **Aim:** Here, we focus on the chemical structures of several arylpiperazine derivatives, highlighting their anti-proliferative activity in different tumor cell lines. The modular structure, diverse biological activities, and potential for combination therapies of arylpiperazine compounds make them valuable candidates for further preclinical and clinical investigations in the fight against cancer. **Conclusion:** This review, providing a careful analysis of different arylpiperazines and their biological applications, allows researchers to refine the chemical structures to improve potency, selectivity, and pharmacokinetic properties, thus advancing their therapeutic potential in oncology.

## 1. Introduction

*N*-arylpiperazines are a class of molecules known to possess antihistamine, anti-inflammatory, and antihypertensive activities. They represent a fundamental scaffold of pharmaceutical chemistry and are the basis of several drugs, especially those involved in the treatment of neurodegenerative diseases. This is why, in recent years, interest in the synthesis of *N*-arylpiperazine derivatives has increased and is currently growing [1]. 

Most of these compounds have a flexible aliphatic chain that can vary in length, linking the arylpiperazine fragment to the second terminal pharmacophore group. 

Piperazines are therefore considered important and biologically active elements; they are scaffolds consisting of a six-term ring and two nitrogen atoms placed at opposite ends of the ring. Structure–activity relationship studies have enhanced their pharmacokinetic properties and, since this moiety is involved in the structure of numerous drugs, their role in various pathways is known [2]. 

In this context, the purpose of this review is to discuss the impact and the potential of the *N*-arylpiperazine scaffold, and our attention is focused on representative examples reported in the literature with the aim to highlight the importance of these molecules and their use in cancer. 

Today, cancer is one of the most feared and life-threatening diseases; consequently, there is growing emphasis among medicinal chemists on developing innovative anticancer agents and refining treatment strategies to target cancer more precisely. Obviously, the main goal is to achieve highly selective targeting on cancer cells so as to radically decrease toxicity on non-transformed cells. 

The limitations in this regard and the various aspects to be improved in cancer therapy are varied: first, the mortality associated with both the disease and the toxicity of the drugs used; then the bioavailability, half-life, and adverse effects of the drugs which are not always on the patient’s side; and finally, the poor quality of life to be led. 

Presently, interest is progressively being directed toward small molecules, and the advances made so far have increasingly helped screening for molecules that have some affinity for tumor receptors [3]. Obviously, a challenge to overcome always remains anticancer drug resistance linked to several mechanisms that often add up and for which multi-drug combination therapy is preferred; in this regard, once again, the efficacy of small molecules has been proven even in this case, although research is always moving forward [4]. 

In this review, we shed more light on the recent literature and the most inspiring studies demonstrating the efficacy and especially the potential of arylpiperazine molecules in cancer. By focusing on the efficacy of these future drugs, their structure, and tumor localization, our goal is to place increasing attention on this promising and ever-present scaffold. 

## 2. The Evidence of *N*-Arylpiperazine Derivatives in Carcinogenic Pathways

Arylpiperazine derivatives are crucial for a variety of biological targets, particularly central nervous system receptors. Indeed, in the literature, their involvement in the regulation of the central nervous system is linked to their activity as possible agonists or antagonists of various serotonergic receptors [5,6]. 

This explains why *N*-1-substituted *N*-arylpiperazines (so-called “long-chain arylpiperazines”) have been thoroughly studied as a structural motif in the design of analogues for this type of receptor; in particular, this moiety is the most extensively studied class of 5-HT_1A_ receptor ligands for serotonin (5-HT) receptors. 

The pharmacophore of serotoninergic receptor agonists is characterized by an aromatic ring and a basic planar nitrogen, and it has been proven that there are two main interactions responsible for the affinity of *N*-arylpiperazine with 5-HT_1A_ receptors (Figure 1): on the one hand, the ionic bond between the protonated nitrogen atom of the piperazine ring and the carboxylic oxygen of the side chain of Asp3.32; on the other hand, an edge-to-face CH–π interaction between the aromatic ring and the Phe6.52 residue [7].

Over the last few years, several novel compounds that target 5-HT_1A_ receptors have advanced into phase II and phase III clinical trials or have already been released commercially as anxiolytics. Specifically, significant effort has been dedicated to understanding the role of the terminal component in the interaction between ligands and receptors. 

Consequently, a wide variety of different fragments have been employed. An example is Buspirone, an anxiolytic from which other analogues have subsequently been derived [8]. Another example of an *N*-arylpiperazine derivative is Flibanserin, an ineffective antidepressant used for hypoactive sexual desire disorder (HSDD). Agonist 5-HT_1A_ and antagonist 5-HT_2A_ have activity on pyramidal neurons in the prefrontal cortex and can enhance dopamine (DA) and norepinephrine (NE) activity and reduce 5-HT activity in many brain regions. This results in improved symptoms of HSDD [9]. 

To better emphasize the importance and potential of this scaffold, Ikwu et al. have performed a study based on a Quantitative Structure Activity Relationship (QSAR) model to design and predict the cytotoxic activity of arylpiperazine derivatives against LNCAP prostate cancer cell line. They explored their molecular docking interaction with the androgen receptor (LNCaP cells have been reported to be androgen-sensitive and depend on androgen for growth). This has indicated that some of these compounds are potent, and their properties are comparable to those of some drugs that are used for prostate cancer [10]. 

The mechanisms related to carcinogenesis are numerous, but one of the purposes of this review is precisely to link arylpiperazines to cancer and try to investigate mechanisms related to proliferation. Among the most widely expressed receptors in different types of cancer is the serotoninergic receptor 5-HT_1A_ (Table 1) [11].

One of the mechanisms by which serotonin regulates proliferation is represented by MAPK/ERK and PI3K/Akt: the activation of serotonin receptors stimulates the molecules involved in this pathway (ERK1/2, Akt, NF-κB) (Figure 2). 

Further studies have also highlighted apoptosis resulting from downstream activation of PLCβ, Ras, and Raf-1 after stimulation of these receptors [12,13,14]. First, it is widely known that prostate cancer is closely related to a number of neuroendocrine cells that release serotonin and a high concentration of 5-HT_1A_ receptors have been found in various prostate cancer cell lines (PC3, DU145, LNCaP) [15]. To prove this, there are several 5-HT_1A_ antagonists, for example NAN-190 (Figure 3), containing an arylpiperazinic scaffold, or Pindobind, which have been shown to inhibit cell proliferation in vitro. 

Similarly, there is evidence that serotonin is involved in bladder cancer and in small-cell lung carcinoma. The involvement of serotonergic receptors in the latter has been proven using antagonists such as SDZ 216–525 (Figure 4). 

In addition, serotonin has demonstrated its mitogenic role in colorectal cancer, as the use of antagonists or SSRIs (such as BW501C or Citalopram and Fluoxetine) has reduced tumoral growth [16]. In this case, having to consider the gastrointestinal tract, several receptor subtypes such as 5-HT_3_, 5-HT_4_, and 5-HT_1_ receptors for colon cancer were studied. Additionally, the proliferative effects of selective receptor agonists (BP554) (Figure 5) on HT29 cells were confirmed [17]. 

Another aspect to be considered as a starting point for future studies is the involvement of this scaffold in the design of selective derivatives of α1A and α1D receptor subtypes. An example could be represented by the drug repurposing of Naftopidil (Figure 6), used for benign prostatic hyperplasia management. Naftopidil, an arylpiperazine-based α1-AR antagonist with a naphthalene group, has demonstrated its potential anticancer activities related to its pharmacological profile in numerous studies [18]. This compound, used in Japan in the treatment of BPH, has emerged as a potential anticancer drug both because it is useful in arresting prostate cell growth but also in decreasing the cell viability of various cell lines such as bladder or renal cell lines. In fact, many research groups have been involved in the development of derivatives tested for their ability to block α1-ARs (specifically α1A, α1B, and α1D) [19,20]. 

There is a lot of evidence showing the antitumor potential of α1 blockers. There are many relevant examples in the literature of antagonists such as Prazosin (used for the treatment of hypertension) or Terazosin (Figure 7) (used in cases of hypertension or urinary symptoms due to benign prostatic hypertrophy); among the mechanisms highlighted are DNA damage stress induction for the former and cell growth inhibition for the latter [21]. 

Similarly, the antitumor effect of Doxazosin (Figure 8), an antihypertensive drug used to treat benign prostatic hyperplasia, has been further investigated. This drug has the arylpiperazine scaffold in its structure and its antiproliferative effects have been demonstrated in several cell lines. It could act by different mechanisms such as the activation of TGF and IkB, the inhibition of PKB/AKT activation and angiogenesis, or by autophagy and Suzuki et al. proved that Doxazosin sensitizes different tumor cells to Osimertinib, a tyrosine kinase inhibitor [22].

Additionally, androgens involved in normal prostate development and also in prostate cancer act through the androgen receptor (AR). These receptors are highly expressed in prostate cancer cells; in fact, they have long been studied as tumor targets for drug development. Evidence of this includes AR-antagonist drugs such as flutamide, hydroxyflutamide, bicalutamide, and also arylpiperazine derivatives (Figure 9) that have demonstrated efficacy on this pathway. Moreover, Qi et al. explored and discussed the SAR of the arylpiperazine derivatives. The lead compound, containing a phenyl ring substituted at the 4-position of the piperazine ring, compared with derivatives supporting a benzyl or a pyridine group at the same position, showed strong cytotoxic activity against LNCaP cells (IC50 = 3.67 µM), and a potent antagonistic potency against AR. These data showed that other aryl groups substituted at the same position of the piperazine ring did not improve activity. Instead, concerning the presence of substituets on phenyl ring, SAR studies indicated that the o-substituted phenyl group derivatives displayed moderate to strong cytotoxic activities against LNCaP cells and relatively strong antagonistic potency against AR [23].

Over all, the androgenic role in male physiological development and the disorders associated with it is well known. Another example is the compound YM-92088 (Figure 10), with high AR antagonist activity, which is even more potent than bicalutamide (IC_50_ value of 0.89 μM) with an IC_50_ value of 0.47 μM [24]. 

### 2.1. N-Arylpiperazine Derivatives in Prostate Cancer

Prostate cancer is one of the most common cancers in the world, counting about 1–3 million new cases annually [25]. Considering cases with or without PSA screening, it is the cause of death in 1–2% of the male population [26]. 

In recent studies, Hong Chen et al. have presented a library of naftodipil-based arylpiperazine derivatives. These novel hybrids have been synthesized, characterized, and evaluated against prostate cancer cell lines (PC-3, LNCaP, and DU145) and their cytotoxicity has been compared with the effects of these compounds in non-cancer human prostate cells WPMY-1. Many of these compounds have exhibited significant cytotoxic activities against LNCaP cells and DU145 cells (more active even than naftodipil and finasteride against DU145 cells), low cytotoxic profiles toward WPMY-1, and have shown α1-Ars selectivity. Compounds **9** and **10** (Table 2) have been evaluated for their effects on cell cycle progression and the result is that compound **10** has greatly increased the number of DU145 cells in the G0/G1 phase (Figure 11), unlike compound **9**.

Always considering the treatment of benign prostatic hyperplasia, prior research indicates that arylpiperazine derivatives could potentially act as α1a and/or α1a- + α1d- selective ligands. Furthermore, they have undertaken additional assessments to explore antagonistic effects utilizing dual-luciferase reporter assays. Their goal was to identify potential subselective antagonist candidates aimed at treating benign prostatic hyperplasia (BPH), among arylpiperazine derivatives recognized for their potent anticancer properties. Finally, compounds **9** and **10** demonstrated heightened selectivity towards specific subtypes of α1-Ars and 10 [27]. Compounds **9** and **10** characterized by an electron-donating group and an electron-withdrawing group on the para position of the phenyl ring showed no different effects on DU145 cell survival. Therefore, the mechanisms of **9** and **10** in inhibiting cancer cells growth are not involved with apoptosis.

Kinoyama et al. have described the synthesis of a series of *N*-arylpiperazine derivatives and the correlated results as these compounds were then subjected to an assessment concerning their androgen antagonist profile. This choice was obviously made because many antiandrogens are currently in use for the treatment of prostate cancer. They focused on modifying compound YM-92088, **11** (Table 3) while considering the piperazine scaffold. The resulting data show that both nitrogen atoms in the piperazine ring are essential for potency. The modifications on one of the compounds led to the synthesis of compound **12** that demonstrated the strongest antiandrogenic activity. Consequently, it was observed that introduction of an additional fluorine atom at the *ortho* position on the phenyl ring may be important for in vivo activity [28].

### 2.2. N-Arylpiperazine Derivatives in Colorectal Cancer

Colorectal cancer is one of the most aggressive forms of cancer; targeted therapy offers a novel approach that has shown promise in significantly prolonging the survival of patients. The number of deaths associated with this type of cancer is considerable even though it has decreased through early screening [29]. 

Szczuka et al. have focused their attention on the role of HSPA1 and HSP90AA1, whose levels turn out to be increased in cancerous colorectal lesions. In fact, the expression of HSPA1 and HSP90AA1, key heat shock proteins involved in facilitating neoplastic transformation and cancer development, is already altered in precancerous colorectal lesions and surrounding tissue, to a degree that is dependent on polyp’s potential for malignancy. Consequently, the effect of Piroxicam, Meloxicam, and new arylpiperazine analogues, previously synthesized as analgesic without any ulcerogenic activity [30,31], was tested on the expression of the already mentioned heat shock proteins in colorectal adenocarcinoma lines (HCT 116, Caco-2, and HT-29 cells). These classic drugs have repeatedly shown anticancer properties, acting through both COX-dependent and independent pathways. Nevertheless, the precise molecular mechanisms behind these effects remain to be fully elucidated. The following compounds (Figure 12) first showed reduced cytotoxicity when compared with the corresponding oxicam, and then showed the ability to differently decrease the protein expression of HSPA1 in all cells under examination. In addition, the oxicam analogues exhibited effectiveness in downregulating the expression of HSP90AA1, a trait not observed in classic drugs. 

In terms of chemical structure, all examined analogues deviate from conventional drugs due to substitutions of the arylpiperazine pharmacophore and benzoyl moiety at the thiazine ring. Indeed, the introduction of arylpiperazine pharmacophore due to high electron-withdrawing properties enhanced the anti-inflammatory properties of the compounds compared with the parent drugs. Moreover, the presence of fluorine on this pharmacophore further enhanced the electron-withdrawing effect and thus the anti-inflammatory properties of the compounds with a 3-carbon propylene linker between nitrogen atoms of thiazine and piperazine rings [31]. 

### 2.3. N-Arylpiperazine Derivatives in Pancreatic Cancer

Pancreatic cancer, most of the time, is diagnosed at an advanced stage or already metastatic given the hard-to-diagnose early stage [32]. Research directed toward screening and new therapeutic strategies can only help in reducing the mortality of this fatal malignancy [33]. 

Through different approaches, Hong Su et al. [34] have explored new strategies for treating pancreatic cancer, as existing ones lead to short-term survival. First, they have considered Sunitinib (SUN), which is generally used to treat different types of cancer and their focus was to examinate the combination of SUN and an arylpiperazine derivative, compound C2 (**18**, Figure 13), which represents a D1DR agonist that was demonstrated as being able to reduce the cancer stem-like cell (CSC) frequency in both pancreatic cells and accordingly enhance the response to SUN in the treatment of pancreatic cancer.

The discovery of new D1DR agonists that are chemically stable and available orally is essential. This is because several studies have shown that dopamine reduces the presence of cancer stem-like cells (CSC), closely associated with the progression, metastasis, and recurrence of pancreatic cancer. Indeed, gradual attention has been paid to the use of chemotherapy and targeting CSCs in the treatment of cancer. The deepened role of dopamine [35,36,37] in the reduction of CSC intratumor and D1DR as a target in anticancer therapy has also been considered and CSC frequency inhibition by compound C2 in pancreatic cancer cells PANC-1 and SW1990 has been confirmed. Additionally, it was found that this compound increases the cell level of cAMP in SW1990 xenograft (level that generally is increased by use of D1DR agonists or by D2DR antagonists), indicating, by molecular docking studies, the higher propensity for D1DR binding than D2DR. C2 compounds supporting an *N*-arylpiperazine moiety could be considered potentially useful in the treatment of pancreatic cancer while also improving the response to Sunitinib [34].

### 2.4. N-Arylpiperazine Derivatives in Breast Cancer

Different type of breast cancer (BC) are heterogeneous and classified according to subtype and corresponding therapy, and despite the reaserch efforts and increasing knowledge, BC represents the most common one along with lung cancer [38].

Also in this context, the role of arylpierazine derivatives as serotonergic ligands emerges as having the potential to target serotonin and connective tissue growth factor (CTGF) signaling, while also ameliorating the sensitivity to Tamoxifen in ER+ breast cancer cells. CTGF has been identified as a glucose-induced modulator of cell sensitivity to tamoxifen and the CTGF silencing induced a significant increase in tamoxifen sensitivity of BC cells grown in hyperglicemia, at levels like those obtained for cells cultured in normal levels of glycemia. The arylpiperazine derivatives (Figure 14) improve the efficacy of tamoxifen on MCF7 breast cancer cells (ER+) by modulating the expression of CTGF. In fact, the N′-cyanopicolinamidine derivative (compound **19**), characterized by the bis (4-fluorophenyl) methyl piperazine moiety, showed affinity in the nanomolar range towards 5-HT_2C_ receptors (Ki = 21.4 nM) and weak or no affinity towards 5-HT_2A_ and 5-HT_1A_ receptors, respectively. Moreover, another interesting derivative (compound **20**), which showed high affinity (Ki value of 1.13 nM) and selectivity toward 5-HT_2C_ receptors, was characterized by the 3,4-dichlorophenyl group as an N-4 piperazine substituent, linked through an ethyl chain to a norbornene fragment [39,40,41].

Successively, a new arylpiperazine scaffold supporting a dihydrothiazole moiety in different positions on the phenyl ring and with the meta position being the most favorable was designed and synthesized by Andreozzi et al. The choice of thiazole ring was made considering that this nucleus was already reported as showing a wide range of pharmacological activities like anti-inflammatory, anti-tubercular, anti-diabetic, anti-malarial, and anticancer [42,43]. The synthesized compounds were subjected to binding assays on 5-HT_1A_ receptors and pharmacological evaluation on breast and prostate cancer cells. Compared with prostate data, all thiazolinylphenyl–piperazine compounds showed a 50% reduction in breast cell viability with a concentration of at least 25 μM. The most interesting finding of this work concerns the result of the **21–23** compounds on the MCF-7 cell line (Table 4): it was evidenced that the acetylated derivatives (**24–26**) did not show any activity on this cell line, suggesting that the antitumor effect was not influenced by increased lipophilicity and consequently cell-permeability.

In addition, a highly cytotoxic effect was observed on MDA-MB231 for both **21–23** and **24–26** acetylated derivatives simultaneously showing a significant selectivity towards non-transformed cells [42]. 

Therefore, given the absence of therapeutic approaches devoid of cytotoxic effects, these results obtained on androgen-independent prostate cancer and triple negative breast cancer cells highlight the potential innovation that these compounds can represent in combating extremely aggressive forms of tumors.

### 2.5. N-Arylpiperazine Derivatives in Cervical Carcinoma

Cervical cancer is one of the most common causes of death in women [44]. Cervical neoplasia begins as an intraepithelial alteration, and generally requires many years to progress into an invasive disease [45]. 

Mao et al. [46], in the context of arylpiperazine derivatives, focused on hybridization for the synthesis of new derivatives and then tested the obtained compounds in vitro for their anticancer activities. These new hybrid compounds have been tested on several cell lines: lung carcinoma (A549), cervical carcinoma (Hela), breast carcinoma (MCF-7), and gastric carcinoma (SGC7901). As shown in Table 5, preliminary anticancer activity of the reported compounds has been proven. From these data, however, it emerged that the compounds with the chlorine or trifluoromethyl substituents on the benzene ring are those that show most cytotoxic activity. So, they focused on just one compound, and they proved that this compound **27** exerts cytotoxic activity selectively against Hela. 

### 2.6. N-Arylpiperazine Derivatives in Leukemia

Considerable efforts have been made over the years to classify the different forms of hematopoietic neoplastic diseases and their respective treatment, but in some cases it continues to represent a fatal malignancy. Some of these, called chronic, develop slowly, leading to high levels of circulating white blood cells. Acute leukemias are characterized by an early growth of white blood cells, and the disease is usually lethal in a short time [47]. Choi et al. [48] have synthesized many aryloxazole derivatives containing an arylpiperazine moiety and acting as vascular-targeting anticancer agents. The most interesting aspect is the dual effect of these compounds, the tumor vasculature disruption, and mitotic arrest. Cytotoxic effects were studied considering human leukemia cells (HL-60), and a careful analysis was made on the importance of substituents and functional groups in this scaffold. In fact, replacing the arylpiperazine group with other heterocycles, they noticed a loss of cytotoxicity. This indicates that the piperazine nitrogen atoms and the substituted aryl group are critical for binding. The vascular-disrupting effect was then tested on selected derivatives that displayed low IC_50_ values. Finally, to confirm the dual effect, they also demonstrated the ability of these compounds to inhibit tubulin polymerization during mitosis. 

Therefore, considering all these data, in vivo studies were conducted selecting compounds that showed the best profiles in inhibition tests of growth cells. On these bases, it was verified that compound **28** (Table 6) is a potential anticancer agent [48] with an outstanding microsomal stability. This compound inhibited tubulin polymerization at low concentrations, suggesting that its biological activity comes from tubulin binding. Moreover, compound **28** showed an excellent antimitotic effect and vascular-disrupting activity in vitro and demonstrated promising antitumor activity in vivo, possibly because of its metabolic stability.

### 2.7. N-Arylpiperazine Derivatives in Melanoma

The incidence of melanoma cases has been increasing lately, especially in fair-skinned countries [49], and the mortality rate is dramatically high [50]. 

Romagnoli et al. [51] have synthesized Cinnamic acid derivatives linked to arylpiperazine moieties with the aim of investigating the role of the enzyme tyrosinase in the melanogenic process, which is one of the most studied therapeutic targets for melanoma because it regulates melanin synthesis. Through structure–activity reports, they studied the influence of the substituent on the arylpiperazine scaffold, and the synthesized derivatives were tested by evaluating the inhibitory effect using mushroom tyrosinase, also evaluating cell viability and tyrosinase activity in A375 human melanoma cells. Finally, toxicity was evaluated by zebrafish assays considering depigmenting effects on zebrafish embryos. It was found that derivative **29** (Table 7) reduces melanogenesis without any toxicity effects up to 100 μM and at a 5-fold lower concentration (20 μM) significantly decreases (60%) the activity of human tyrosinase in α-melanocortin (MSH)-stimulated A375 cells.

### 2.8. Other Examples of Arylpiperazines in Cancer

Finally, there are many promising arylpiperazines that have been tested for their activity as anticancer agents on different cell lines. 

Lee et al. evaluated quinoxalinyl–piperazine derivatives as possible anticancer agents [52]. Among them, they identified one as a growth inhibitor of cancer cells and specifically demonstrated that this compound is a G2/M-specific cell cycle inhibitor and inhibits anti-apoptotic Bcl-2 protein with p21 induction. Compound **30** (Figure 15) inhibited the proliferation of several lines of cancer cells, including breast, skin, pancreas, and cervix cells, as shown in Table 8. This inhibition is dose dependent. 

In addition, the association between this compound and other anticancer drugs (Figure 15), such as taxanes, paclitaxel, a platinum derivative cisplatin, a topoisomerase II selective agent doxorubicin, gemcitabine, and fluorouracil, has led to a synergistic inhibitory effect.

Moreover, some potent smoothened receptor (SMO) inhibitors, implicated in the carcinogenesis process, are piperazine-based compounds (Figure 16). Among them, ANTA XV (**31**) reduced tumor growth when administered in a genetic mouse model of medulloblastoma, with regression observed at the two highest doses administered [53]. Similarly, NVP-LEQ506 (**32**) has been developed and activity has been tested against an SMO mutant cell line (D473H) identified in a patient with medulloblastoma who had relapsed after an initial response to vismodegib [54].

Another arylpiperazine-based compound, SGI7079 (**33**, Figure 17), was found to be a potent selective Axl inhibitor, the effect of which manifests itself in a dose-dependent manner with an IC50 of 58 nM. Indeed, the overexpression of AXL has been found in numerous cancer types, which most often correlates with a worse patient prognosis. Thus, by inhibiting the kinase, these arylpiperazine derivatives should be able to overcome the resistance of EGFR inhibitors in some cancers [55]. 

## 3. Conclusions

Over the years, interest in *N*-arylpiperazine derivatives in the field of pharmaceutical chemistry has been growing. At the beginning, derivatives were mainly studied as serotonergic ligands, finding application in the development of drugs active on the central nervous system (CNS). Subsequently, due to the chemical–physical characteristics of this basic scaffold, studies were undertaken on different pharmacological models including some molecular targets involved in carcinogenesis and tumor progression. In this context, many examples are reported in the literature that have shed more light on the potential of the arylpiperazine moiety useful in the development of new antitumor agents. 

Notably, these findings have also translated into the market entry of some arylpiperazine-based drugs, such as Imatinib (**34**), Dasatinib (**35**), Palbociclib (**36**), Milciclib (**37**), Rociletinib (**38**), and Abemaciclib (**39**) (Figure 18), as targeted therapies approved by FDA [56,57,58] for the treatment of cancer. These results, together with the current studies described in this review, clearly emphasize the role of arylpiperazine derivatives not only on the CNS but also in cancer.

Consequently, research in this area requires further efforts to clarify the molecular target and the transduction mechanism of inhibition of the pathways involved in the various tumor forms, and highlight the activity of the described molecules. Moreover, it is also desirable that in the field of medicinal chemistry more interest could be directed towards the design and synthesis of new chemical entities including arylpiperazine-based molecules. To this purpose, to investigate all the different portions of this scaffold and the different substituents on the aromatic ring, it could be interesting to perform SAR studies that lead to the development of novel compounds characterized by increased selectivity against cancer cell lines and reduced toxicity. This last issue is of particular interest since a synergistic use in combination with classic chemotherapeutics could be hypothesized for these molecules. In this way, conventional anticancer therapy would be used with lower doses and consequently have fewer toxic effects.

In conclusion, the promising results presented in this review, and the presence of *N*-arylpiperazine moiety in several FDA-approved anticancer medications, makes it a desirable scaffold with great potential for the development of novel anticancer drugs.

## Figures and Tables

**Figure 1 pharmaceuticals-17-01320-f001:**
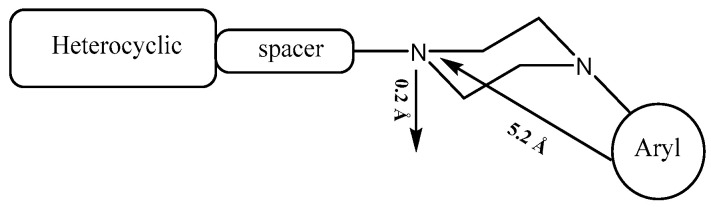
General structure of *N*-arylpiperazine and pharmacophoric model of 5-HT_1A_ agonist.

**Figure 2 pharmaceuticals-17-01320-f002:**
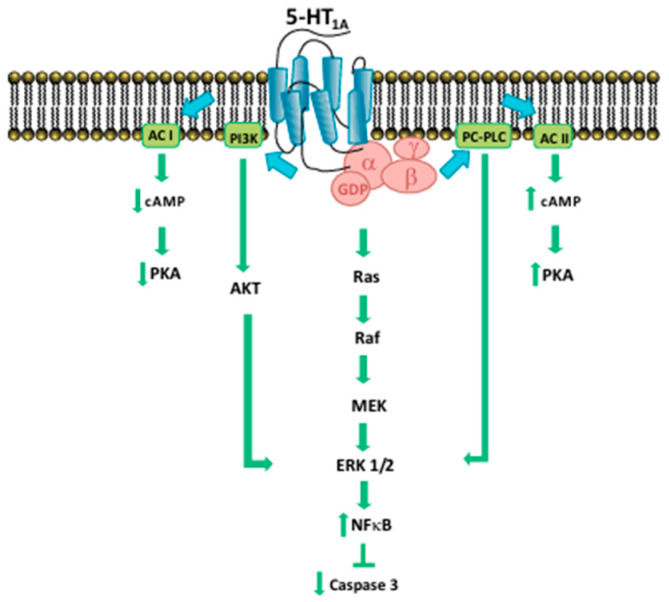
Signaling pathways of 5-HT_1A_ receptor [11].

**Figure 3 pharmaceuticals-17-01320-f003:**
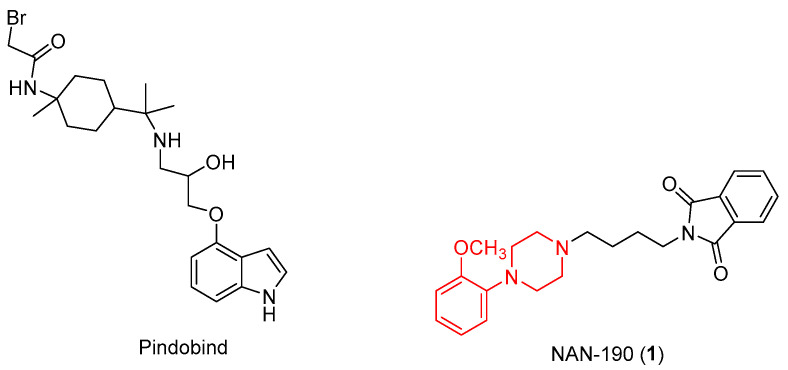
Chemical structure of5HT_1A_ antagonists.

**Figure 4 pharmaceuticals-17-01320-f004:**
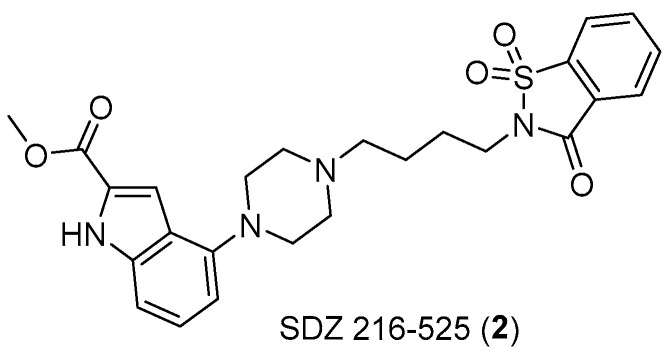
Structure of SDZ 216–525 (**2**).

**Figure 5 pharmaceuticals-17-01320-f005:**
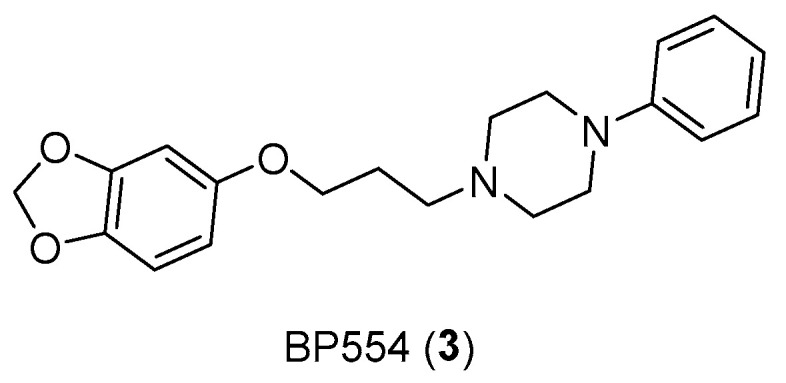
Structure of BP554 (**3**).

**Figure 6 pharmaceuticals-17-01320-f006:**
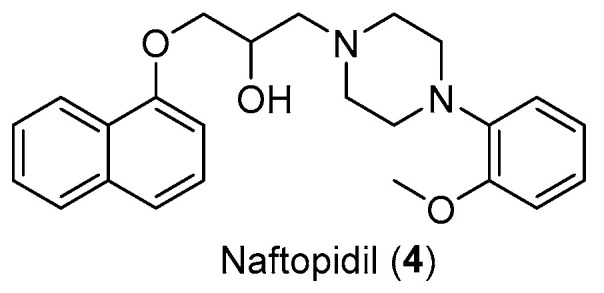
Structure of Naftopidil (**4**).

**Figure 7 pharmaceuticals-17-01320-f007:**
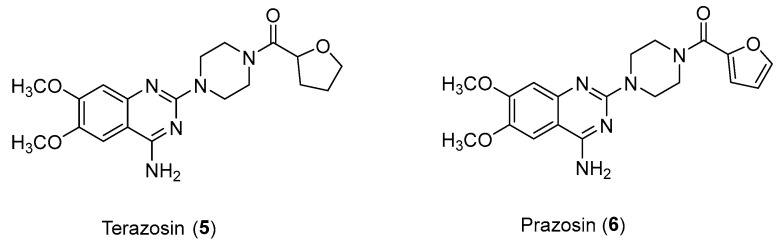
Structures of Terazosin (**5**) and Prazosin (**6**).

**Figure 8 pharmaceuticals-17-01320-f008:**
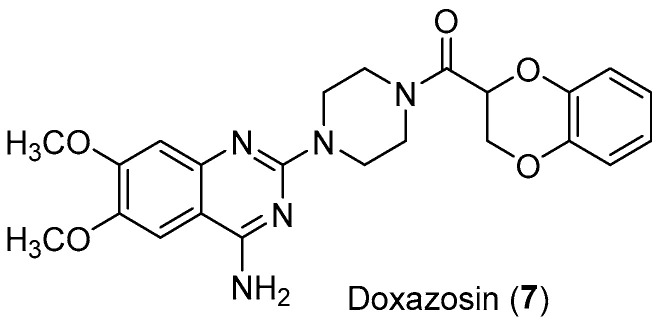
Structure of Doxazosin (**7**).

**Figure 9 pharmaceuticals-17-01320-f009:**
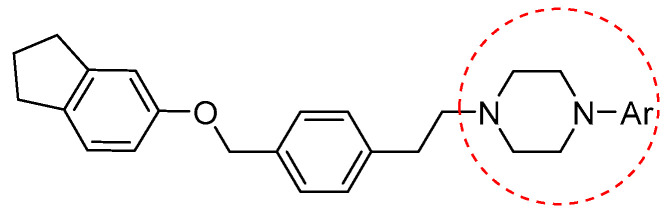
General structure of arylpiperazine derivatives. Ar indicates variously substituted phenyl or benzyl piperazine.

**Figure 10 pharmaceuticals-17-01320-f010:**
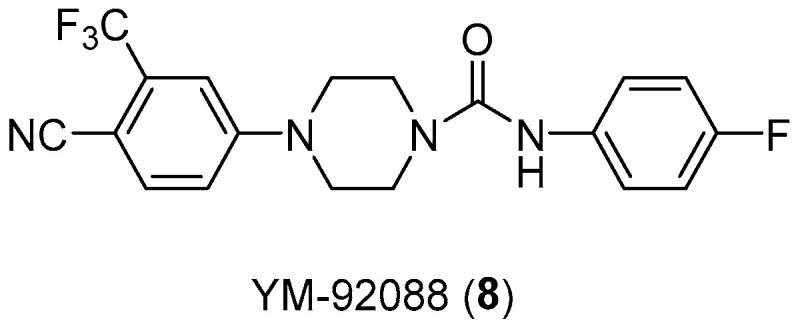
Structure of androgen antagonist YM-92088 (**8**).

**Figure 11 pharmaceuticals-17-01320-f011:**
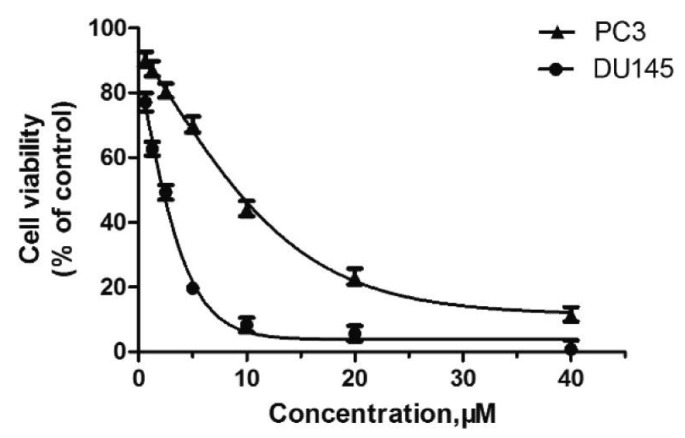
Compound **10** inhibited cell viability in prostate cell lines PC-3 and DU145. All cells were exposed to escalating concentrations of arylpiperazine derivatives respectively for 24 h, and the cell viability was detected by CCK-8 assay [27].

**Figure 12 pharmaceuticals-17-01320-f012:**
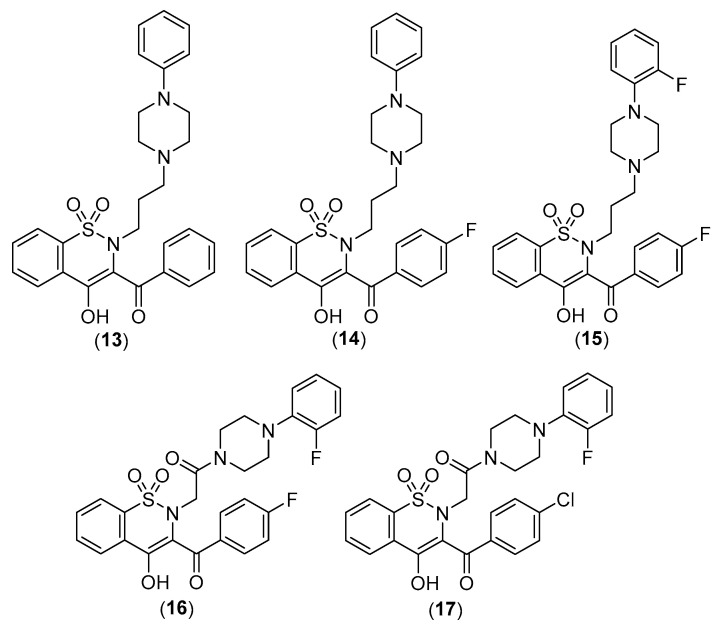
Chemical structures of oxicams derivatives acting on colorectal adenocarcinoma cell lines.

**Figure 13 pharmaceuticals-17-01320-f013:**
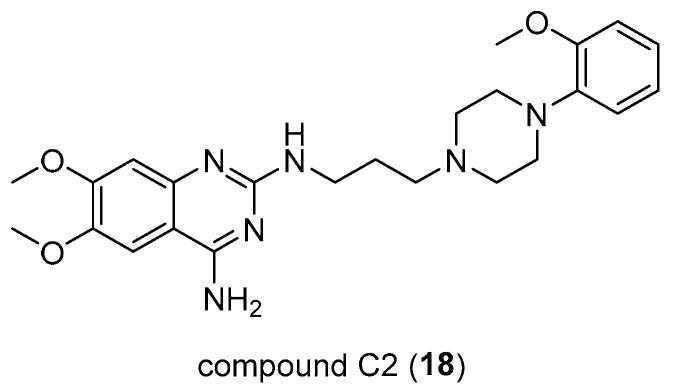
Chemical structure of C2 (**18**).

**Figure 14 pharmaceuticals-17-01320-f014:**
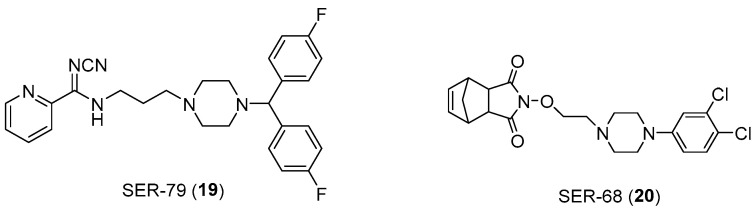
Chemical structure of SER-79 (**19**) and SER-68 (**20**).

**Figure 15 pharmaceuticals-17-01320-f015:**
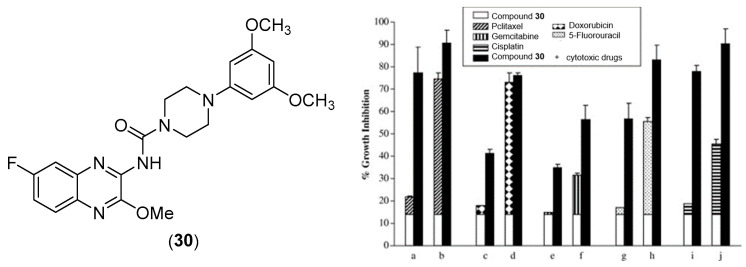
Chemical structure of compound **30** and its combination effect with different cytotoxic drugs on the growth of MDA-MB-231 cancer cells [52]. The drugs were used at the following doses: (a,b): 2 and 3 nM paclitaxel; (c,d): 5 and 10 nM doxorubicin; (e,f): 100 and 200 nM gemcitabine; (g,h) 3 and 5 μM fluorouracil; (i,j): 300 and 500 μM cisplatin.

**Figure 16 pharmaceuticals-17-01320-f016:**
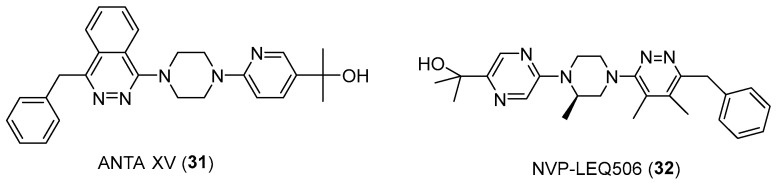
Chemical structures of SMO inhibitors.

**Figure 17 pharmaceuticals-17-01320-f017:**
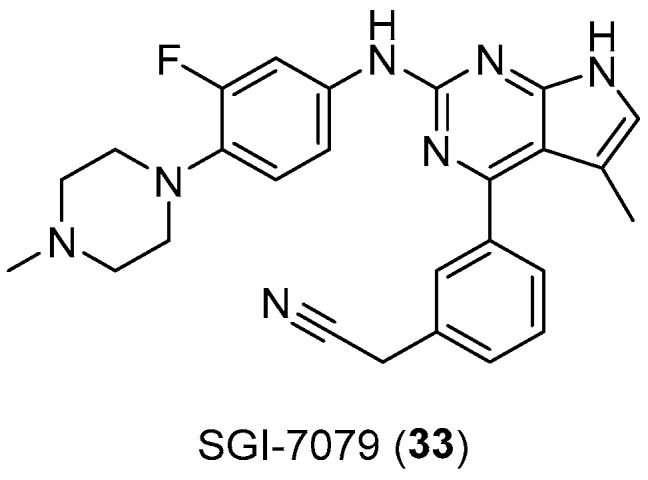
Chemical structure of SGI-7079 (**33**).

**Figure 18 pharmaceuticals-17-01320-f018:**
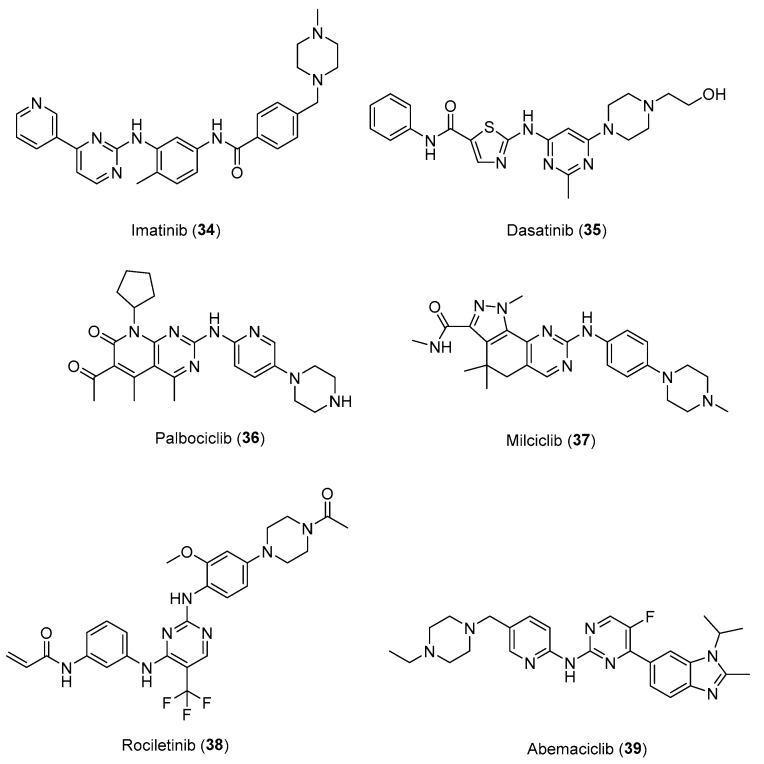
Potent anticancer drugs bearing *N*-arylpiperazine moieties.

**Table 1 pharmaceuticals-17-01320-t001:** The expression of 5-HT_1A_ receptor in cancer cells.

Type of Cancer	Cell Lines Expressing 5-HT_1A_R	Drugs	Effects
Prostate Cancer	PC3, DU145, LNCap	NAN-190 Pindobind	5HT_1A_ antagonists that inhibit cell growth in vitro, inducing apoptosis.
		6-nitroquipazine Zimelidine Fluoxetine	5HT uptake inhibitors that cause dose-dependent inhibition of cells proliferation.
Bladder Carcinoma	SHT1376	NAN 190 SB224289	5HT_1A_ (NAN-190) and 5HT_1B_ (SB224289) antagonists that show an inhibitory effect on the serotonin-induced growth cells.
Small Cell Lung Carcinoma	GLC8	Spiperone SDZ 216–525	5-HT_1A_ (spiperone) and 5-HT_7_ (SDZ 216–525) antagonists that inhibit 8-OH-DPAT-induced mitogenic effect.
Colorectal Carcinoma	HT29	BW501C Citalopram Fluoxetine	Serotoninergic antagonists (BW501C) and SSRIs (Citalopram and Fluoxetine) that retard the tumor growth.
		NAN 190 SB224289	5HT_1A_ (NAN-190) and 5HT_1B_ (SB224289) antagonists that reduce cell growth acting as antiproliferative agents.
Cholangiocarcinoma	Mz-chA1, HuH28, HUVV-T1, CCLP-1, SG231, TFK1.	-	-

**Table 2 pharmaceuticals-17-01320-t002:** Compound **9** and **10**: data of selectivity ratio and IC_50_ values.

Compd.	Structure	Selectivity Ratio	IC_50_ Values (DU145)
**9**	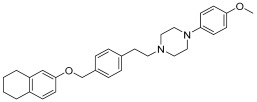	α1B/α1 Aratio = 16.7	0.93 ± 0.19
**10**	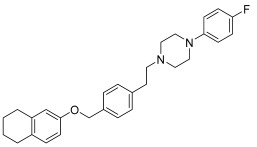	α1B/α1D ratio = 10.9	0.90 ± 0.20

**Table 3 pharmaceuticals-17-01320-t003:** Compounds **11** and **12**: AR antagonistic activities.

Compd.	Structure	IC_50_ (μM) ^a^	% inhibition ^b^
**YM-92088** **(11)**	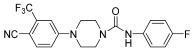	0.47	-
**12**	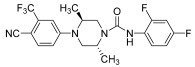	0.20	85% ** ED_50_ = 1.1 mg/kg

^a^ compound was tested for their ability to inhibit AR-mediated transcriptional activation using a reporter assay. IC_50_ values were determined by a single experimental run-in triplicate. ^b^ The mean percent changes from the respective control value of ventral prostate weight after oral administration in testosterone propionate-treated castrated rats (10 mg/kg/d for 11 d, n 5 or 6). ** *p* 0.01 versus control by Dunnett’s multiple comparison test.

**Table 4 pharmaceuticals-17-01320-t004:** IC_50_ values (μM) of novel thiazolinylphenyl-piperazines (**21–26**) observed on breast cancer cell lines.

Compound	Structure	MCF-7	MDA-MB231
**21**	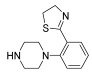	14.70 ± 1.9	31.37 ± 5.1
**22**	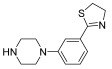	15.93 ± 1.8	39.96 ± 9.8
**23**	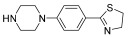	19.47 ± 2.3	36.32 ± 7.7
**24**	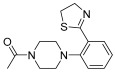	-	23.27 ± 3.4
**25**	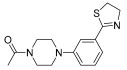	-	34.60 ± 5.4
**26**	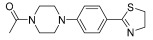	-	47.15 ± 6.7

**Table 5 pharmaceuticals-17-01320-t005:** In vitro cytotoxic activity of compound **27**.

Compound 27	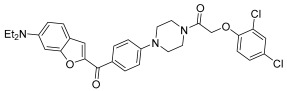		
Cell Lines (IC_50_ μM)			
A549	Hela	MCF-7	SGC7901
5.73 ± 1.22	0.03 ± 0.04	12.38 ± 3.62	6.17 ± 1.62

**Table 6 pharmaceuticals-17-01320-t006:** Cytotoxic effect of 6–48 derivative (**28**) against Human Leukemia Cells (HL-60).

Compound	Structure	IC_50_ (nM)
**28**	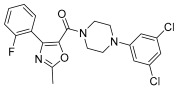	60.2

**Table 7 pharmaceuticals-17-01320-t007:** Effect of compound **29** on the diphenolase activity of mushroom tyrosinase.

Compound	Structure	IC_50_ (μM)
**29**	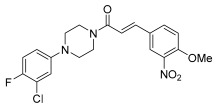	0.51 ± 0.10

**Table 8 pharmaceuticals-17-01320-t008:** Inhibition of cell growth (IC_50_, μM) by quinoxalinyl–piperazine compound **30** against human cancer cell lines.

Human Cancer Cell Lines	IC_50_, μM
MDA-MB-231	0.012
Caki-1	0.011
UMRC2	0.013
PANC-1	0.021
A549	0.021
MKN-45	0.020
HepG2	0.019
HCT116	0.020
HT29	0.021
PC-3	0.021
U251	0.015
HeLa	0.021
SK-MEL-28	0.020
OVCAR-3	0.012

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
