# Peer review of "Arylpiperazine Derivatives and Cancer: A New Challenge in Medicinal Chemistry"

_pharmaceuticals, 2024, doi:10.3390/ph17101320_

Round 1
Reviewer 1 Report
Comments and Suggestions for Authors
Thorough English language revision is required
Reviewer 2 Report
Comments and Suggestions for Authors
The chosen field of review is interesting and opens up a promising direction in medical chemistry, however, significant revision of the text on data presentation is necessary. The presentation of materials requires improvement. Usage data in illustrations without references to experimental work is incorrect.
Reviewer 3 Report
Comments and Suggestions for Authors
The manuscript entitled “Arylpiperazine derivatives and cancer: a new challenge in medicinal chemistry” submitted by Fiorino et al, is an interesting review that gathers valuable information for researchers in medicinal chemistry focused on the design of new antineoplastic drugs containing this important scaffold in their structure.
However, this manuscript needs to be completed and corrected at different points, to be accepted for publication.
Some major revisions are:
1. Some paragraphs need to be rearranged so as not to repeat information and to give it an order. For example: in the introduction, the reference to certain ligands of the 5-HT1A receptor should be included in point 2, as one of the many antitumor compounds that act in certain carcinogenic pathways. Thus, the background information contained in point 1 should point to general considerations of the arylpiperazine fragment and its relevance in the design of new antineoplastic drugs. Therefore, I believe that from line 40 to 75 should be moved to a subsection of point 2. The rest of the paragraphs in section 1 are in the same vein and are consistent with an introduction.
Consider that the paragraph starting on line 76 and the one on line 97 point to the same thing and should be made into one.
2. In section 2, following the previous suggestion, it could be subdivided into the signaling pathways mentioned therein related to carcinogenic processes.
2.1 Signaling pathway of 5-HT receptor (different subtypes mentioned)
2.2 Adrenergic receptors...
2.3 Androgen receptor...
In this part it is very important to include Bcr-Abl oncoprotein present in patients suffering from chronic myeloid leukemia and in some other types of leukemia (ALL), and that its aberrant activity activates the MAPK/ERK/AKT signaling pathway, especially because the drugs and analogous compounds with important inhibitory activity of this kinase, possess in their structure the arylpiperazine fragment, such as dasatinib.
3. Therefore, it would be advisable to review in the literature other examples of arylpiperazines that are not incorporated, such as various SMO antagonists and their role in medulloblastomas (Anta XV and NVP-LEQ-506), CDK2 inhibitors (milciclib) or Axl inhibitors (SGI-7079), which would provide more background to those described and which should be included in a review.
4. For the sake of better presentation of the information, there should be a point 3 referring to examples of arylpiperazine derivatives in certain types of cancer.
5. Replace references 1 and 2 with references that are more appropriate.
Some minor revisions:
1. Remove the full stop in the title.
2. In line 23 of the abstract the word ‘Alzheimer’ appears, I think it does not make sense here.
3. Capitalise the first letters of figures throughout the text, as well as the table.
4. In Table 1, correct “Colonrectal” cancer.
5. Unify the names of certain drugs Pindobind (line 117) or piroxicam (line 232).
6. Correct, polyp potential.... on line 231.
7. Bold compound numbers throughout the manuscript.
Round 2
Reviewer 1 Report
Comments and Suggestions for Authors
Thank yo for having done the corrections needed.
Reviewer 3 Report
Comments and Suggestions for Authors
The authors considered all the corrections made in order to have a completer and more useful version for the scientific audience. Therefore, this manuscript should be published.